# AgentExpt: Automating AI Experiment Design with LLM-based Resource Retrieval Agent

Yu Li [1]   Lehui Li [2]   Lin Chen [3]   Qingmin Liao [1]   Fengli Xu [1]   Yong Li [1]

## Abstract

In modern AI research, baseline and dataset selection is a high-stakes decision in experimental design. It operationalizes a research idea into a concrete evaluation protocol and largely determines the validity and comparability of empirical conclusions. However, making appropriate choices is increasingly difficult as baselines and datasets proliferate, while suitability is inherently context-dependent and rarely captured by baseline and dataset metadata. To address these challenges, we present **AgentExpt**, a comprehensive framework for baseline and dataset recommendation. We first curate a large-scale, high-quality knowledge base that links 108,825 accepted papers to their used baselines and datasets. Based on this resource, we design a *collective perception-enhanced retriever* that represents each baseline or dataset by integrating first-person self-descriptions with third-person citation contexts, thereby effectively positioning them within the scholarly network. We further design a *reasoning-augmented reranker* that encodes baseline-dataset interaction chains as a reasoning prior to fine-tune an LLM, producing refined rankings with interpretable justifications. Experiments show that our framework outperforms the strongest baseline, with average gains of +5.85% in Recall@20 and +7.90% in HitRate@10, and ablation studies confirm the effectiveness of our designed components. Overall, AgentExpt advances the efficient and reliable automation of experimental design. Our code is available at https://github.com/tsinghua-fib-lab/AgentExpt.

[1]Tsinghua University, Beijing, China [2]Shandong University, Jinan, China [3]Northeastern University, Boston, USA. Correspondence to: Fengli Xu <fenglixu@tsinghua.edu.cn>, Yong Li <liyong07@tsinghua.edu.cn>.

*Proceedings of the 43rd International Conference on Machine Learning*, Seoul, South Korea. PMLR 306, 2026. Copyright 2026 by the author(s).

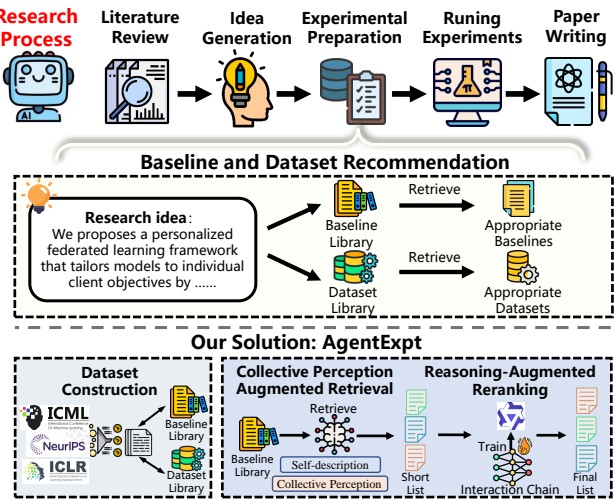

*Figure 1.* Overview of the research problem and our proposed solution.

## 1. Introduction

Baseline and dataset selection is a recurring bottleneck in modern AI research. Beyond being a clerical choice, it determines the evaluation target, the comparison standard, and the data distribution under which claims are tested, thereby shaping whether results are comparable, reproducible, and scientifically meaningful (Torralba & Efros, 2011). This step has become increasingly difficult as the ecosystem of baselines and datasets expands and fragments across tasks, versions, and evaluation conventions (Ott et al., 2022). Crucially, experimental suitability is context-dependent: a baseline or dataset that is appropriate in one setting may be misleading or uninformative in another (Paullada et al., 2021), yet such suitability cues are rarely explicit in item metadata or portal descriptions (Figure 1).

In parallel, recent progress in large language models (LLMs) and agentic systems has spurred interest in automated research workflows (Brown et al., 2020; Radford et al., 2019; 2018; Shang et al., 2024b;a; Li et al., 2025; Zhao et al., 2025), ranging from literature review and idea generation (Huang et al., 2025; Team et al., 2025; Li et al., 2026) to broader pipeline automation (Institute, 2025; Intology, 2025; Lu et al., 2024; Yamada et al., 2025). However, most existing systems focus on upstream stages or operate in con-

trolled experimental settings (e.g., fixed benchmarks and predefined tasks), and therefore treat baseline and dataset choices as given rather than as a decision variable. As a result, reliable automation of baseline and dataset recommendation remains comparatively underexplored despite being central to rigorous experimental design.

Three limitations in existing research remain salient. First, **coverage is limited**: many systems draw candidates from domain-restricted collections or portal-curated inventories (e.g., Paper With Code), which do not reliably reflect what is actually used across the literature and therefore miss a substantial portion of relevant baselines and datasets (Ivanova et al., 2023; Viswanathan et al., 2023; Färber & Leisinger, 2021a). Second, **suitability is under-modeled**: even when using supervised text classification (Färber & Leisinger, 2021b;a), dense retrieval (Viswanathan et al., 2023), graph-based models (Altaf et al., 2019; Qayyum et al., 2025), or collaborative filtering (Ivanova et al., 2023), most approaches rely on first-person descriptions and metadata, effectively treating suitability as topical similarity and overlooking how artifacts are positioned and used in practice. Third, **baseline selection and dataset selection are typically decoupled**, and thus recommendations ignore their joint compatibility and the fact that evaluation intent is often realized through baseline-dataset pairings rather than either component in isolation.

In contrast, day-to-day research practice typically starts from a small set of closely related papers and follows their trails to the baselines and datasets they actually used, rather than browsing portal listings; researchers also consider baselines and datasets jointly, favoring pairings that have been used together or are methodologically compatible with the intended setting. Moreover, experimental choices must be justified by situating selected baselines and datasets relative to prior work, making citation contexts a valuable third-person signal that complements first-person self-descriptions. Together, these observations motivate a unified formulation that links papers, baselines, and datasets and explicitly leverages community-level perception.

Motivated by these observations, we present *AgentExpt*, a comprehensive framework for baseline and dataset recommendation that directly targets the three limitations above (Figure 1). To address limited **coverage**, we curate a large-scale, literature-grounded knowledge base that links papers to the baselines and datasets they actually use. Concretely, we build an automated data-collection pipeline that gathers 108,825 accepted papers from the past decade across ten flagship AI venues, parses the experiment sections from PDFs, and employs LLM-assisted extraction to identify baseline and dataset mentions. We then perform entity normalization and source linking to resolve naming variations and map extracted mentions to canonical components, yield-

ing a high-precision paper–component linkage that reflects real usage in experiments rather than portal listings.

To model **context-dependent suitability** beyond metadata-only similarity, we design a *collective perception-enhanced retriever* that augments first-person self-descriptions with third-person evidence from the literature (Figure 4). Specifically, for each baseline or dataset, we extract its citation contexts from downstream papers as community-level perception signals, summarize these contexts into an aggregate usage profile, and integrate it with artifact's self-description to form a dual-view representation. We then fine-tune an embedding model on these representations to efficiently retrieve candidates whose usage patterns, not only topical wording, align with the evaluation intent implied by a query.

Finally, to capture the **synergy** between baselines and datasets, we design a *reasoning-augmented reranker* that leverages paper-mediated interaction evidence between baselines and datasets to construct structured reasoning traces. This module extracts paper-baseline-paper-dataset (P-B-P-D) and paper-dataset-paper-baseline (P-D-P-B) interaction chains, and uses them to construct reasoning chains. We use these traces as a reasoning prior to fine-tune an LLM for reranking, enabling refined recommendations accompanied by interpretable justifications that are both semantically grounded and aligned with evaluation intent.

We evaluate *AgentExpt* on the curated knowledge base against four families of baselines. Compared with the strongest baseline, *AgentExpt* yields average gains (across the two tasks) of **+5.85%** in Recall@20, **+8.30%** in HitRate@5, and **+7.90%** in HitRate@10. Ablations show that collective perception contributes the largest share of retrieval gains, while interaction-chain reasoning drives consistent improvements in hit rates.

We summarize the key contributions as follows:

- We present *AgentExpt*, a comprehensive framework that automates baseline and dataset recommendation for experimental design under context-dependent suitability.

- We curate a large-scale, high-quality knowledge base that links 108,825 papers to the baselines and datasets they use, enabling realistic training and evaluation for *AgentExpt* and future research.

- We design a *collective perception-enhanced retriever*, integrating first-person self-descriptions with third-person citation contexts to capture community usage signals beyond metadata-only matching.

- We design a *reasoning-augmented reranker*, encoding baseline-dataset interaction chains as a reasoning prior to fine-tune an LLM for refined rankings with interpretable justifications.

- We empirically validate the state-of-the-art performance of *AgentExpt*, and the effectiveness of each component.

## 2. Related Work

### 2.1. LLM-Based Agents for Research Assistance

LLM-based agents have increasingly automated scientific workflows, ranging from idea generation (Baek et al., 2024; Pu et al., 2025; Team et al., 2025) and algorithm optimization (Novikov et al., 2025; Fawzi & Paredes, 2023) to end-to-end paper production (Lu et al., 2024; Yamada et al., 2025; Intology, 2025; Institute, 2025; Tang et al., 2025; Schmidgall et al., 2025; Huang et al., 2025). However, these systems generally bypass detailed experimental design, failing to systematically recommend appropriate baselines and datasets essential for rigorous evaluation. In contrast, we introduce a framework to automate baseline and dataset retrieval.

### 2.2. Datasets for Baseline and Dataset Recommendation

Early search engines lacked scientific context awareness (Brickley et al., 2019; Chapman et al., 2020), while the only paper-conditioned baseline recommendation dataset, RecBaselines2023 (Ivanova et al., 2023), is limited in scale and restricted to recommender systems. For dataset recommendation, approaches have evolved from fixed-label classification (Färber & Leisinger, 2021b) to open-domain retrieval (Viswanathan et al., 2023). However, these datasets typically source candidates from public portals (e.g., Wikidata, Papers with Code) rather than actual literature, resulting in significant coverage gaps regarding artifacts used in published experiments. In contrast, we construct a comprehensive dataset by extracting baselines and datasets directly from papers across ten flagship AI venues, ensuring high coverage of real-world research practices.

### 2.3. Methods for Baseline and Dataset Recommendation

Prior approaches span supervised classification to fixed labels (Färber & Leisinger, 2021b;a), graph-based learning on heterogeneous networks (Qayyum et al., 2025; Altaf et al., 2019), and dense neural retrieval (Viswanathan et al., 2023), alongside collaborative filtering strategies for baselines (Ivanova et al., 2023). However, these methods typically prioritize surface-level semantic similarity or structural co-occurrence, often overlooking experimental suitability and failing to provide interpretable justifications for their choices. In contrast, we propose a framework that integrates collective citation perception to capture usage intent and leverages interaction-chain reasoning to deliver recommendations that are both experimentally suitable and interpretable.

*Table 1.* Comparison of datasets for recommending baselines or datasets.

| Dataset | # Papers | # Baselines | # Datasets | Top conf | Repo links |
|---|---|---|---|---|---|
| **AgentExpt** | 108,825 | 116,970 | 68,316 | ✓ | ✓ |
| DataFinder | 17,495 | N/A | 7,000 | ✗ | ✗ |
| RD4SPD | 88,000 | N/A | 1,413 | ✗ | ✗ |
| RecBaselines | 903 | 363 | N/A | ✗ | ✗ |

## 3. Preliminary

### 3.1. Problem Formulation

We formalize baseline and dataset recommendation as a retrieval task in the context of scientific research. Given a research idea expressed as a natural language query $q$, the objective is to retrieve from a large candidate pool the most suitable subset of experimental components, including baselines and datasets, that are appropriate for evaluating the proposed idea.

Let $\mathcal{Q}$ denote the space of research ideas, $\mathcal{B}$ the inventory of available baselines, and $\mathcal{D}$ the inventory of datasets. For a given query $q \in \mathcal{Q}$, the system should return two subsets:

$$R_B(q) \subseteq \mathcal{B}, \quad R_D(q) \subseteq \mathcal{D},$$

where $R_B(q)$ and $R_D(q)$ represent the recommended baselines and datasets respectively. These outputs are intended to approximate the ground-truth relevance sets

$$G_B(q), \quad G_D(q),$$

Unlike generic document retrieval, this task requires reasoning not only about topical similarity but also about experimental suitability.

### 3.2. Input Representation

We instantiate the query $q$ using the abstract summary of a paper that expresses the research idea. Formally, for a paper $p$ with abstract $a(p)$, we set

$$q = \mathrm{Summ}\big(a(p)\big),$$

where $\mathrm{Summ}(\cdot)$ is implemented with GPT-4o (OpenAI, 2024) to produce a concise synopsis that preserves the problem setting, assumptions, and evaluation intent. This abstract based representation captures the key scientific context.

### 3.3. Evaluation Metrics

Following conventions in information retrieval, we evaluate recommendation performance by comparing the retrieved subsets $(R_B(q), R_D(q))$ against the gold sets $(G_B(q), G_D(q))$. The primary metrics are:

- **Recall@$k$**: The fraction of gold items (baselines or datasets) that appear in the top-$k$ retrieved results. High

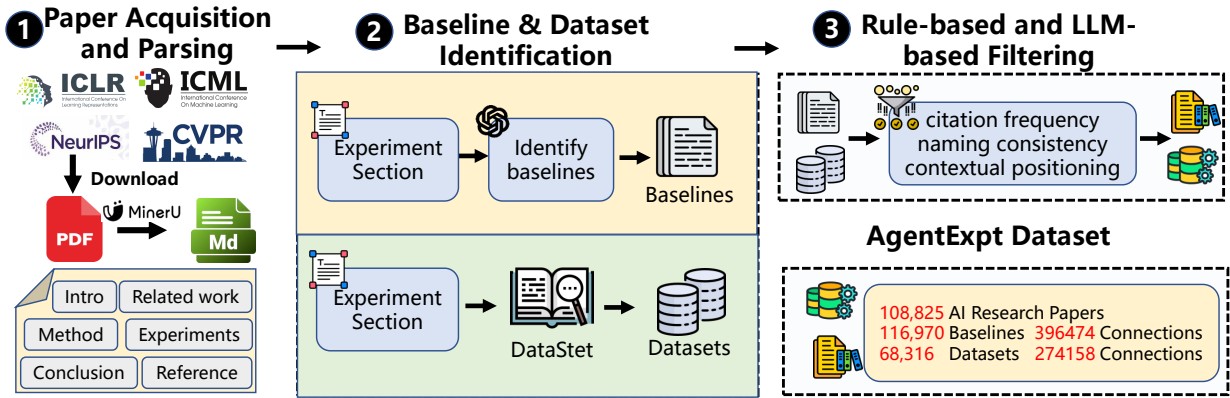

*Figure 2.* Pipeline for constructing the AGENTEXPT knowledge base. We (1) download and parse papers (from flagship AI conferences), (2) identify baselines and datasets by locating experiment sections, (3) apply rule-based and LLM-based filtering using citation frequency, naming consistency, and contextual positioning to prune false positives. The final dataset contains 108,825 papers, 116,970 baseline entities, 68,316 dataset entities, and their respective cross-entity connections.

recall ensures that critical experimental components are not missed.

- **HitRate@$k$**: A binary measure that equals 1 if at least one relevant item is retrieved within the top-$k$, and 0 otherwise. This reflects a user-centric view of success: whether system surfaces some useful component early.

## 4. Dataset Construction

### 4.1. Overview

To effectively support the retrieval and recommendation of state-of-the-art baselines and suitable evaluation datasets corresponding to specific research topics, it is imperative to precisely extract baselines and datasets utilized in empirical evaluations reported in scholarly literature. Accomplishing this task requires extensive extraction of structured entities accompanied by comprehensive metadata associated with respective papers, baselines, and datasets. However, current academic resources (e.g., Paper With Code, HuggingFace) and automated information extraction methodologies (Viswanathan et al., 2023; Färber & Leisinger, 2021a)) exhibit substantial limitations concerning dataset coverage, entity metadata completeness, and the accuracy of identifying *Paper-Baseline*, *Paper-Dataset* associations.

Specifically, establishing a robust and high-quality Paper-Baseline-Dataset data involves addressing two primary challenges. **First**, significant *heterogeneity* in data sources and formats complicates information extraction efforts. Baselines often appear in abbreviated or inconsistent terminologies across different publications, while datasets frequently suffer from ambiguous naming conventions and inconsistent references. Moreover, the absence of explicit links within papers to original baseline implementations and dataset repositories further complicates automated text parsing, entity recognition, and information alignment processes. **Second**, the exponential growth of research outputs exacerbates in-

efficiencies and scalability constraints inherent to manual or semi-automated annotation methods. Leading AI conferences receive tens of thousands of submissions annually, making traditional annotation approaches impractical and highlighting the need for scalable automation.

Motivated by recent advancements in automated scientific information extraction (Viswanathan et al., 2023), we design a comprehensive, end-to-end automated pipeline for scholarly data extraction and validation. Our pipeline specifically targets research papers published between 2015 and 2025 across ten prominent AI conferences: NeurIPS, ICML, ICLR, ACL, NAACL, EMNLP, CVPR, ICCV, AAAI, and IJCAI, because they collectively represent flagship venues spanning major AI subfields with consistently high review standards, providing a literature-grounded and high-quality source for extracting experimental configurations at scale.

### 4.2. Automated Dataset Construction Pipeline

To address the challenges of data heterogeneity and the scalability limitations of manual annotation, we design an end-to-end automated pipeline. Specifically, the pipeline comprises three critical stages: *Acquisition and Parsing*, *Baseline and Dataset Identification*, and *Rule-based and LLM-based Filtering*, see Figure 2.

**Acquisition and Parsing.** We systematically acquire targeted research papers and their corresponding metadata from prominent AI academic repositories, including OpenReview, ArXiv, and official conference websites. The collected metadata includes comprehensive information such as paper titles, authors, abstracts, publication years, conference affiliations, and direct links to publications. After rigorous de-duplication and cleaning, the corpus exceeding 108,825 papers was refined into a high-quality dataset suitable for subsequent analyses. Utilizing the *MinerU* tool (Wang et al., 2024), we perform structured parsing on each paper, accu-

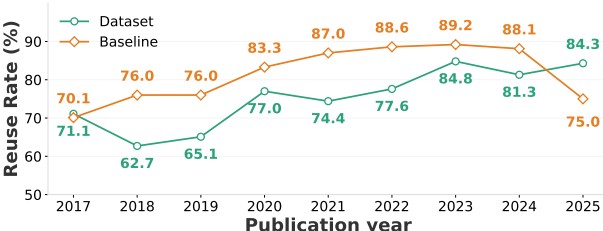

*Figure 3.* The reuse rate of experiment baselines and datasets. The vertical axis indicates the reuse rate, calculated as the fraction of resources employed in year N that were introduced in preceding years (1 to N-1), reflecting the dependency on established experimental components over time.

rately extracting chapter structures, paragraph boundaries, table arrangements, and citation anchors, thereby ensuring reliable downstream entity extraction.

**Baseline and Dataset Identification.**   To identify baselines and datasets used in each research, we resort to citations within experiment-centric sections (e.g., *Experiments*, *Results*, *Evaluation*) as anchors. For baseline identification, we analyze the citation context combined with additional metadata, such as introductory papers and repository links, that are collected from resources like Paper With Code and OpenAlex (Priem et al., 2022). For dataset identification, we use the *DataStet* (Lopez, 2024) model to automatically extract dataset entities from the experimental sections, validating them through metadata from platforms like Hugging-Face and Kaggle. This process includes entity normalization, alias resolution, and filtering, ensuring high-quality and accurate data for subsequent validation stages.

**Rule-based and LLM-based Filtering.**   To further enhance extraction accuracy and effectively mitigate false-positive candidates, we introduce a two-stage filtering strategy. Initially, we employ rule-based filtering mechanisms to preliminarily evaluate candidates by leveraging criteria such as contextual positioning, citation frequency, and naming consistency. Subsequently, we introduce an additional verification stage for entities near the decision boundary using few-shot prompting with GPT-4o. Specifically, we implement a unified context-aware evaluation protocol where the LLM integrates extracted textual evidence with external metadata to perform entity normalization and rigorous verification, ensuring high-confidence identification for both baselines and datasets.

### 4.3. Data Analysis

We conduct an extensive quality assessment and statistical analysis of the curated knowledge base. It comprises 116,970 distinct baseline methods and 68,316 evaluation datasets used by 108,825 research papers, establishing approximately 326,697 paper-dataset links and 543,205 paper-baseline links. Additionally, the completeness ratio of repository links achieves 13%.

A key finding from our analysis is the remarkably high reuse rate of established resources within the community. As Figure 3 shows, both baselines and datasets exhibit strong recurring adoption patterns over time. For instance, the baseline reuse rate consistently remains above 70% in recent years, with several periods even exceeding the 85% threshold. This indicates that our dataset successfully captures the core set of canonical methods and benchmarks that form the foundation for experimental comparisons in AI research.

These results underscore that our automated extraction and validation pipeline notably surpasses existing methodologies in terms of informational completeness and its ability to map the central, actively-used research components. Consequently, our approach substantially enhances dataset scale, information richness, and breadth of coverage, offering more reliable and comprehensive support for academic recommendation systems and reproducibility initiatives.

## 5. Method

We design two modules for dataset and baseline recommendation. (1) *Collective perception-augmented retrieval*: We represent each candidate by concatenating its self-description with collective perception, and finetune an embedding model on these representations to improve recall of components. (2) *Reasoning-Augmented Reranking*: We extract paper-baseline–paper-dataset (paper-dataset–paper-baseline) interaction chains from the literature and train a language model to generate reasoning chains over these interactions to rerank the shortlist.

### 5.1. Collective Perception-Augmented Retrieval

We build retrieval representations that jointly encode a candidate's first-person self-description and third-person collective perception distilled from citation contexts, and then finetune a dense retriever to maximize recall of baselines and datasets, as shown in Figure 4.

**Citation Context Extraction**   Extracting faithful citation contexts is non-trivial. Mentions of a target $t$ (baseline or dataset) are often ambiguous (acronyms and aliases) and non-indicative of use (generic related-work prose) with multiple targets sometimes packed into one sentence. Naively collecting surrounding sentences therefore injects noise and dilutes the signal about how $t$ is actually positioned and used. To reduce this noise, given a corpus paper $p$, we first locate experiment-centric sections. For each mention of a target $t$ (baseline or dataset) in $p$, we extract a localized window around the mention consisting of the containing sentence

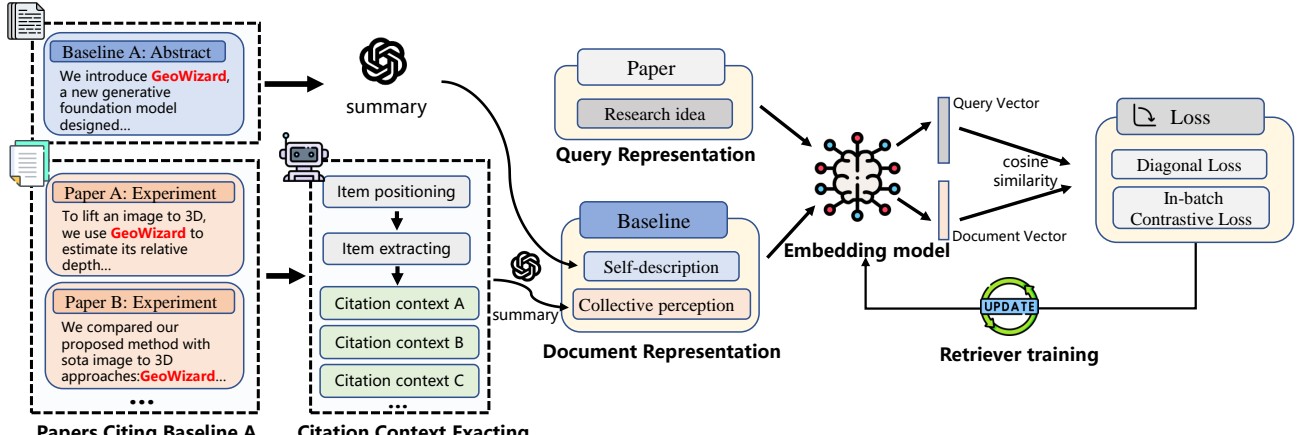

*Figure 4.* Illustration of Collective Perception Augmented Retrieval

plus its immediate neighbors to capture the stated purpose, setting, and outcome of using $t$. We keep these windows as raw citation contexts without further post-processing. This yields a set $C_p(t) = \{c_1, \ldots, c_m\}$ for $p$. Because a given target $t$ may be used across many papers, we aggregate all contexts over the corpus as $C(t) = \bigcup_p C_p(t)$, which provides multiple, potentially diverse, evidence snippets about when and how $t$ is used in practice.

**Collective Perception** Our goal is to capture a target's collective perception. A naive choice is to treat the pooled citation contexts $C(t)$ as this signal. We avoid this because a popular target may accrue thousands of mentions, far beyond the embedding model's context window. To address these issues, We instantiate a target's Collective Perception (CP) as a citation context summary. For each target $t$, we use GPT-4o (OpenAI, 2024) to synthesize $\text{CP}(t)$ from its pooled citation contexts $C(t)$. The summary answers when and why the baseline or dataset is used, distilling recurring signals about task assumptions, evaluation protocols (metrics, splits, scales), typical configurations, and common co-usage patterns. During synthesis, we de-duplicate paraphrases, discard background or non-experimental mentions, and enforce an evidence-grounded prompt that favors majority-supported observations and avoids speculation.

**Target Representation** We adopt early text-level fusion by concatenating the target's self-description with its collective perception $\text{CP}(t)$:

$$x_t = [\texttt{Description}(t)] \parallel [\texttt{CP}(t)].$$

This single-pass encoding lets the encoder's global self-attention align fine-grained signals across the two segments.

**Retriever Finetuning** We finetune a Qwen-based bi-encoder retriever (Qwen3-Embedding-0.6B (Zhang et al., 2025)) with contrastive supervision.

*Query and candidate formatting.* For a query paper $p$, we

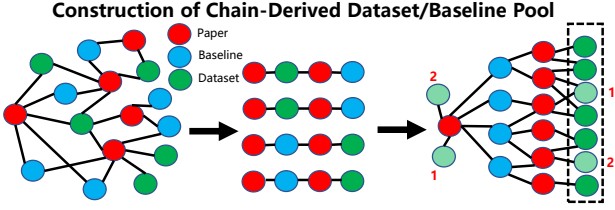

*Figure 5.* **Constructing chain-derived candidates.** From the interaction graph of papers, baselines, and datasets, we extract interaction chains and aggregate the terminal items to form a chain-derived dataset/baseline pool.

form a synopsis $q = \text{Summ}(a(p))$ as in §3. We prepend an instruction prefix to queries only:

$$\texttt{Instruct:} \quad \mathcal{T} \; \texttt{Query:} \quad q,$$

where $\mathcal{T}$ describes the task (e.g., *"Given a research idea, retrieve relevant baseline methods or datasets that are most suitable."*). Candidates remain in their native form without an instruction:

$$x_t = [\texttt{Description}(t)] \parallel [\texttt{CP}(t)].$$

Let $f_\theta(\cdot)$ denote the encoder and $\oplus$ denote string concatenation. We obtain embeddings $h_q = f_\theta(\texttt{Instruct} \oplus q)$ and $h_t = f_\theta(x_t)$, and define the temperature-scaled similarity

$$s(q, t) = \tau \langle h_q, h_t \rangle, \quad \tau > 0.$$

*Loss.* Given a batch $\{(q_i, t_i, y_i)\}_{i=1}^B$ with one aligned positive per row (non-diagonal entries serve as in-batch negatives), we compute the similarity matrix $S_{ij} = \tau \langle h_{q_i}, h_{t_j} \rangle$ and optimize a diagonal binary cross-entropy with an in-batch contrastive regularizer:

$$\mathcal{L} = \underbrace{\frac{1}{B} \sum_{i=1}^B \text{BCE}(S_{ii}, y_i)}_{\text{diagonal (main)}} + \lambda \underbrace{\frac{1}{B(B-1)} \sum_{i \neq j} \text{BCE}(S_{ij}, 0)}_{\text{in-batch contrastive}}.$$

This objective pulls each query toward its paired positive while pushing it away from other candidates in the batch, without explicit hard-negative mining.

*Table 2.* Comparison on paper–baseline and paper–dataset recommendation with conservatively estimated Recall@10/20/30 and Hitrate@5/10/15.

| Method | Paper–Baseline | | | | | | Paper–Dataset | | | | | |
|---|---|---|---|---|---|---|---|---|---|---|---|---|
| | R@10 | R@20 | R@30 | HR@5 | HR@10 | HR@15 | R@10 | R@20 | R@30 | HR@5 | HR@10 | HR@15 |
| BM25 (Robertson et al., 2009) | 0.2080 | 0.2971 | 0.3417 | 0.4239 | 0.5171 | 0.5730 | 0.1334 | 0.1906 | 0.2192 | 0.3143 | 0.4107 | 0.4685 |
| DataHunter (Färber & Leisinger, 2021a) | 0.1809 | 0.3105 | 0.3947 | 0.3913 | 0.4862 | 0.6025 | 0.1302 | 0.2150 | 0.2444 | 0.3091 | 0.4378 | 0.4839 |
| Textual-GCL (Qayyum et al., 2025) | 0.2516 | 0.3594 | 0.4133 | 0.4457 | 0.5696 | 0.6439 | 0.1168 | 0.1668 | 0.1918 | 0.3273 | 0.4309 | 0.4931 |
| SciBERT (Beltagy et al., 2019) | 0.2734 | 0.3905 | 0.4491 | 0.5241 | 0.5965 | 0.6399 | 0.1601 | 0.2287 | 0.2630 | 0.3452 | 0.4674 | 0.5407 |
| DataFinder (Viswanathan et al., 2023) | 0.2889 | 0.4203 | 0.4818 | 0.5284 | 0.6108 | 0.6742 | 0.1857 | 0.2647 | 0.3105 | 0.4059 | 0.4894 | 0.5436 |
| HAtten (Gu et al., 2022) | 0.2918 | 0.4168 | 0.4793 | 0.5360 | 0.6216 | 0.6730 | 0.1803 | 0.2575 | 0.2961 | 0.3912 | 0.4778 | 0.5298 |
| SymTax (Goyal et al., 2024) | 0.2953 | 0.4218 | 0.4851 | 0.5476 | 0.6453 | 0.7039 | 0.2011 | 0.2873 | 0.3304 | 0.4210 | 0.5125 | 0.5674 |
| **AgentExpt** | **0.3422** | **0.4523** | **0.5036** | **0.5933** | **0.6938** | **0.7387** | **0.2236** | **0.3001** | **0.3533** | **0.4557** | **0.5549** | **0.5968** |

## 5.2. Reasoning-Augmented Reranking

This stage constructs reasoning chain representations from observed P-B-P-D/P-D-P-B interactions(see Figure 5) and finetunes a language model to rerank the shortlist with interpretable, chain-based justifications.

**Interaction Chain**  Because baselines and datasets do not form direct citation links, we construct interaction chains centered on papers. We maintain a heterogeneous graph with nodes $\mathcal{V} = \{\text{papers}, \text{baselines}, \text{datasets}\}$ and typed edges: paper→baseline (used-as), paper→dataset (evaluated-on), and co-usage links. Because our extraction is paper-centric, we build chains by expanding small ego-nets around the query paper $p$:

$$p \xrightarrow{\text{dataset}} d \xrightarrow{\text{used-by}} p' \xrightarrow{\text{baseline}} b,$$

which supports baseline recommendation via $p \to d \to p' \to b$. Dataset recommendation is symmetric: $p \to b \to p'' \to d$. We provide a detailed analysis validating the high coverage and selectivity of these chains in Appendix A.5.

**Reasoning Chain**  To unlock LLM reasoning and enable reasoning reranking based on interaction chains, we use GPT-4o to generate reasoning chain trajectories. For each query and each candidate baseline $b$, we enumerate chains of the form $p \to d \to p' \to b$ and score each chain by

$$m_{D \to B}(d, b) = \big|\{\, p' \,:\, p' \text{ uses } d \text{ and } b \,\}\big|$$

We retain only the three chains with the largest $m_{D \to B}(d, b)$ per candidate baseline and expose these chains and their counts to the LLM as the primary evidence. Symmetrically, for each *candidate dataset* $d$, we enumerate $p \to b \to p'' \to d$ and score by

$$m_{B \to D}(b, d) = \big|\{\, p'' \,:\, p'' \text{ uses } b \text{ and } d \,\}\big|$$

again keeping only the top three chains per candidate dataset. A teacher LLM (GPT-4o) is prompted to reason step-by-step and produce a final ordering, with instructions to rely primarily on the top-three chains and their co-usage metrics, thereby yielding interpretable reranking.

**ReRanker Training**  To endow the LLM with interaction-chain reasoning, we finetune it as a listwise reranker via supervised finetuning (SFT). Each training instance is a triplet $(Q, R, A)$: $Q$ is the query, $R$ is the chain evidence for each candidate (top-3 interaction chains with co-usage counts $m_{D \to B}$ or $m_{B \to D}$), and $A$ is the target reasoning chain ending with a normalized decision. The model takes $(Q, R)$ as input and is trained to generate $A$ via token-level negative log-likelihood:

$$\theta^* = \arg\min_\theta \; \mathbb{E}_{(Q,R,A)\sim\mathcal{D}} \Big[ -\sum_{t=1}^{|A|} \log p_\theta(a_t \,|\, a_{<t},\, Q,\, R) \Big].$$

At inference, given $(Q, R)$ the LLM outputs a brief justification and calibrated ranked list, from which we read off the final ranking.

## 6. Experiment

### 6.1. Experimental Setup

**Baselines.**  To strictly evaluate performance, we compare our method against seven representative baselines spanning four distinct paradigms: (i) **Lexical Matching**: BM25 (Robertson et al., 2009); (ii) **Supervised Classification**: DataHunter (Färber & Leisinger, 2021a); (iii) **Dense Retrieval**: SciBERT (Beltagy et al., 2019) and the state-of-the-art retriever DataFinder (Viswanathan et al., 2023); and (iv) **Structure and Graph-aware Methods**: Textual-GCL (Qayyum et al., 2025), HAtten (Gu et al., 2022), and SymTax (Goyal et al., 2024). Detailed descriptions for these baselines are provided in Appendix A.1.

**Implementation Details.**  We evaluate our framework using a rigorous chronological split (8:1:1 for train/val/test) to prevent temporal leakage and simulate real-world usage. For retrieval, we fine-tune `Qwen3-Embedding-0.6B` to recall the top-20 candidates. For the reasoning-augmented reranking stage, we fine-tune `DeepSeek-R1-Distill-Qwen-7B` to reorder these candidates. Further training configurations and hardware specifications are detailed in Appendix A.1.

**Knowledge Base Validation.**  To quantitatively validate the quality of our extracted paper-baseline and paper-dataset links, we conducted a rigorous manual evaluation. We randomly sampled 100 papers from our constructed knowledge

*Table 3.* Ablation study of retriever.

| | Method | R@20 | HR@5 | HR@10 |
|---|---|---|---|---|
| | w/o Collective Perception | 0.3716 | 0.4719 | 0.5691 |
| Baseline | w/o Self-Description | 0.4197 | 0.5349 | 0.6211 |
| | **AgentExpt (Retriever)** | **0.4523** | **0.5717** | **0.6757** |
| | w/o Collective Perception | 0.2493 | 0.3292 | 0.4162 |
| Dataset | w/o Self-Description | 0.2709 | 0.4016 | 0.4978 |
| | **AgentExpt (Retriever)** | **0.3001** | **0.4250** | **0.5308** |

*Table 4.* Ablation study of reranker.

| | Method | HR@5 | HR@10 |
|---|---|---|---|
| | w/o Reasoning Argumented | 0.4284 | 0.5812 |
| Baseline | w/o Interaction Chain | 0.5245 | 0.6355 |
| | **AgentExpt (full)** | **0.5933** | **0.6938** |
| | w/o Reasoning Argumented | 0.3641 | 0.4248 |
| Dataset | w/o Interaction Chain | 0.4126 | 0.5047 |
| | **AgentExpt (full)** | **0.4557** | **0.5549** |

*Table 5.* Ablation study of recommendation pipeline.

| | Method | HR@5 | HR@10 |
|---|---|---|---|
| Baseline | w/o Reranker | 0.5717 | 0.6757 |
| | **AgentExpt (full)** | **0.5933** | **0.6938** |
| Dataset | w/o Reranker | 0.4250 | 0.5308 |
| | **AgentExpt (full)** | **0.4557** | **0.5549** |

base and manually annotated the ground-truth baselines and datasets based on the full text. Comparing our automated pipeline's extracted links against this manual gold standard, our system achieved a precision of 91% and a recall of 95%. These results demonstrate that our rule-based and LLM-assisted filtering pipeline is highly reliable and yields a high-quality experimental resource graph.

## 6.2. Main Results

As detailed in Table 2, AgentExpt consistently outperforms all baseline methods across both recommendation tasks. Compared to the strongest prior competitor, SymTax (Goyal et al., 2024), our framework yields substantial improvements in both retrieval coverage and ranking precision. Specifically, for paper-baseline recommendation, AgentExpt improves Recall@20 by 7.23% and HitRate@10 by 7.52%. Similarly, in the paper-dataset setting, our method surpasses SymTax by 4.46% in Recall@20 and notably by 8.27% in HitRate@10. It is worth highlighting that the improvements in precision-oriented metrics (HitRate) generally exceed those in Recall, particularly for dataset recommendation. This trend validates our two-stage design: while *Collective Perception* ensures a high-quality candidate pool, the *Reasoning-Augmented Reranker* effectively leverages interaction chains to prioritize experimentally suitable components over those that are merely topically similar.

## 6.3. Ablation Study

**Ablation of Retriever.** As Table 3 shows, both collective perception and self-description contribute vital signals to retrieval process. Removing collective perception causes the most substantial performance degradation, with Recall@20 dropping by approximately 17.8% for baselines and 16.9% for datasets. Excluding self-descriptions also leads to a consistent decline, reducing Recall@20 by 7.2% and 9.7% respectively. These findings indicate that while first-person descriptions provide a necessary foundation, the third-person insights derived from community citation contexts are decisive for accurately capturing experimental suitability.

**Ablation of Reranker.** As Table 4 shows, both the interaction chains and the reasoning augmentation contribute significantly to final performance. Removing the reasoning

supervision causes a sharp decline in HitRate@5, dropping by 27.8% for baselines and 20.1% for datasets. Similarly, excluding interaction chains leads to performance degradation, with HitRate@5 falling by 11.6% and 9.5% respectively. These results confirm that high-quality evidence (interaction chains) and the capability to interpret it (reasoning alignment) are both indispensable for accurate reranking.

**Ablation of Recommendation Pipeline.** To assess the impact of the reranking module, we perform an ablation by removing the reranker (denoted w/o Reranker) and comparing its performance to the full model. As Table 5 shows, removing the reranker caused a notable drop in HitRate@5 and HitRate@10 for both the baseline and dataset recommendation tasks. This performance loss shows that while the retriever identifies candidate items, it cannot effectively prioritize the most relevant ones. In contrast, the reranking component, utilizing interaction chains and contextual reasoning, refines rankings and improves both ranking quality and recommendation precision. These results highlight the importance of integrating retrieval and reranking, demonstrating that the reranker significantly enhances the recommendation pipeline's effectiveness.

**Ablation of LLM Dependency.** To verify that our performance gains stem from the proposed architecture rather than the language capabilities of GPT-4o, we conducted two additional ablations. First, we evaluated the system using raw, unedited abstracts as queries instead of GPT-4o generated synopses. Under this setting, AgentExpt still significantly outperforms the strongest baseline, SymTax (which was also evaluated on raw abstracts). Second, we replaced the GPT-4o synthesized Collective Perception (CP) with a naive concatenation of raw citation contexts (Raw CP). As shown in Table 6, AgentExpt with Raw CP suffers negligi-

*Table 6.* Performance robustness without LLM preprocessing. We report results using raw abstracts (no LLM query summarization) and raw CP (no LLM context denoising).

| Task | Method | R@10 | R@20 | R@30 | HR@5 | HR@10 | HR@15 |
|---|---|---|---|---|---|---|---|
| Baseline | SymTax (w/ Raw Abstract) | 0.3116 | 0.4329 | 0.4992 | 0.5687 | 0.6541 | 0.7182 |
| | **AgentExpt (w/ Raw Abstract)** | **0.3601** | **0.4718** | **0.5097** | **0.6014** | **0.7132** | **0.7485** |
| | AgentExpt (w/ Raw CP) | 0.3319 | 0.4434 | 0.4977 | 0.5871 | 0.6809 | 0.7295 |
| | **AgentExpt (w/ GPT-4o CP)** | **0.3422** | **0.4523** | **0.5036** | **0.5933** | **0.6938** | **0.7387** |
| Dataset | SymTax (w/ Raw Abstract) | 0.2089 | 0.3014 | 0.3428 | 0.4344 | 0.5189 | 0.5796 |
| | **AgentExpt (w/ Raw Abstract)** | **0.2458** | **0.3250** | **0.3719** | **0.4851** | **0.5709** | **0.6199** |
| | AgentExpt (w/ Raw CP) | 0.2167 | 0.2937 | 0.3484 | 0.4498 | 0.5518 | 0.5942 |
| | **AgentExpt (w/ GPT-4o CP)** | **0.2236** | **0.3001** | **0.3533** | **0.4557** | **0.5549** | **0.5968** |

ble performance drops and continues to comfortably surpass SymTax. These results confirm that the improvements are fundamentally driven by our mechanism of leveraging third-person community signals and interaction chains, rather than reliance on proprietary LLMs.

### 6.4. Human-Expert Evaluation

To directly validate the practical applicability of our framework and assess recommendation quality beyond mere paper-reconstruction metrics, we conducted a human-expert evaluation. We invited 15 experienced researchers (PhDs and Masters with top-tier publications) spanning five domains: Computer Vision, NLP, Audio, Time-Series, and Optimization (three experts per domain). For 50 randomly sampled papers (10 per domain), evaluators rated the experimental suitability of candidates on a 1-5 Likert scale. We compared the Ground Truth (actual choices made by the original authors) against AgentExpt's Top-3 recommendations (strictly excluding the ground truth).

As shown in Table 7, the unseen baselines and datasets recommended by AgentExpt received high Mean Opinion Scores (MOS) of 4.12 and 4.38, respectively. This demonstrates that even when our system recommends items outside of the original authors' historical choices, domain experts still consider these recommendations to be highly suitable and genuinely reasonable for the intended experimental setting.

*Table 7.* Human-expert evaluation of experimental suitability (1-5 Mean Opinion Score).

| Source of Candidate | Baseline Suitability | Dataset Suitability |
|---|---|---|
| Ground Truth (Actual Usage) | 4.35 | 4.62 |
| AgentExpt (Unseen Top-3) | 4.12 | 4.38 |

## 7. Conclusion

We present *AgentExpt*, a unified framework for recommending both datasets and baselines. We contribute a large-scale knowledge base linking papers to methodological components across ten major AI venues. We formalize collective perception as a community signal and finetune a dense retriever to improve recall. We further introduce an interaction-chain-guided reranker that improves ranking quality while making the decision process transparent. Experiments show consistent gains over strong baselines, highlighting the value of community signals and explicit chain reasoning. More broadly, *AgentExpt* offers community-grounded infrastructure for experimental design by linking literature, experimental components, and supporting evidence. It also enables LLM-based automated research to make experimental choices that are evidence-grounded and auditable, rather than driven by ad-hoc prompting or brittle heuristics.

**Limitations and future work.** Our current resource focuses on major AI venues, and extending coverage to other communities and scientific domains is an important next step. In addition, since the knowledge base is constructed from a time-bounded snapshot of the literature, newly released artifacts after the cutoff may not be immediately captured, motivating incremental updates. Finally, because community signals reflect historical usage and reporting practices, they should be interpreted as supportive evidence rather than normative judgments, and deployment should allow for controlled exploration beyond the most frequent choices.

## Acknowledgments

This work was supported in part by the National Natural Science Foundation of China (Grant No. 62472241) and the National Key Research and Development Program of China (Grant No. 2024YFC3307605).

## Impact Statement

This paper presents AgentExpt, a framework designed to automate the retrieval of baselines and datasets, thereby accelerating the scientific discovery process and lowering the barrier to entry for researchers. By grounding recommendations in large-scale citation data, our work promotes experimental rigor and reproducibility. *AgentExpt* may lower barriers for newcomers and small teams by surfacing widely used evaluation choices with supporting evidence, and it can serve as a grounding layer for LLM-based automated research workflows that require concrete, checkable experimental choices. At the same time, reliance on community signals can reinforce popularity bias, favoring established artifacts and consolidating mainstream directions. To mitigate this risk, recommendations should be treated as decision support rather than prescriptions, and practical deployments should expose evidence, diversify outputs, and allow users to tune exploration versus exploitation.

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

# A. Appendix

## A.1. Experimental Setup.

**Baselines.** We compare the proposed AgentExpt framework against a comprehensive set of baselines categorized into four paradigms. The details of each method are as follows:

- **Lexical Matching**:

  - **BM25** (Robertson et al., 2009): A widely used probabilistic retrieval model based on exact keyword matching and TF-IDF weighting. It serves as a standard baseline to measure surface-level textual relevance without semantic understanding.

- **Supervised Classification**:

  - **DataHunter** (Färber & Leisinger, 2021a): This method treats recommendation as a multi-label text classification problem. It utilizes feature-based classifiers to map natural language problem descriptions to a fixed inventory of dataset labels. Note that as a classification-based method, it is limited to the closed set of datasets seen during training.

- **Dense Retrieval**:

  - **SciBERT** (Beltagy et al., 2019): A BERT model pretrained on a massive corpus of scientific text. We use it as a dense encoder to generate semantic embeddings for both queries and candidates, performing retrieval based on cosine similarity.
  - **DataFinder** (Viswanathan et al., 2023): A state-of-the-art bi-encoder retriever explicitly trained on large-scale dataset-paper pairs with hard negative mining. It represents the strongest baseline among current dense retrieval approaches for scientific data.

- **Structure and Graph-aware Methods**:

  - **HAtten** (Gu et al., 2022): A hierarchical attention network that captures the document structure by attending to words at the sentence level and sentences at the document level.
  - **SymTax** (Goyal et al., 2024): A structure-aware method that leverages syntactic dependencies within the text to improve representation learning for retrieval.
  - **Textual-GCL** (Qayyum et al., 2025): A graph contrastive learning approach that constructs a heterogeneous graph of papers and entities, learning representations by contrasting structural views to capture citation or co-occurrence topology.

**Implementation Details.** To rigorously evaluate our framework under a realistic setting, we adopt a time-based data splitting strategy. Specifically, we sort all collected papers chronologically by publication date and partition them into training, validation, and testing sets with a ratio of 8:1:1. This chronological split strictly prevents temporal information leakage and faithfully simulates the practical scenario where the system recommends established resources for newly drafted papers. In the experimental pipeline, after Stage-1 retrieval, we pass the top-20 candidates to Stage-2 for reasoning-augmented reranking. We utilize `Qwen3-Embedding-0.6B` as the base encoder for retrieval and fine-tune `DeepSeek-R1-Distill-Qwen-7B` as the backbone for the reasoning reranker. All models are trained on a single node equipped with $2\times$ NVIDIA A800 GPUs.

## A.2. Additional Ablation Study: Impact of Retriever and Reranker Choices

To further validate the necessity of our proposed Collective Perception-Enhanced Retriever and the specific efficacy of our Interaction Chain-based Fine-Tuning, we conducted additional ablation studies comparing AgentExpt against two baselines:

- **BM25 + GPT-4o Reranker**: This setup combines a standard lexical retriever (BM25) with a powerful, off-the-shelf LLM (GPT-4o) for reranking. This tests whether a strong general-purpose reasoner can compensate for a weaker retrieval stage.

- **AgentExpt w/o Finetuning**: This setup uses our proposed Citation-Context Augmented Retriever but employs a standard GPT-4o model for reranking without the specific Supervised Fine-Tuning (SFT) on interaction chains. This isolates the contribution of our reasoning-aware SFT strategy.

The results are summarized in Table A.8.

*Table A.8.* Comparison of different retriever-reranker combinations. **AgentExpt (Full)** refers to our proposed method using both the specialized retriever and the interaction-chain finetuned reranker.

| Task | Method | R@20 | HR@10 |
|------|--------|------|-------|
| | BM25 + GPT-4o | 0.2971 | 0.5427 |
| Baseline | AgentExpt (w/o Finetune) | 0.4523 | 0.6501 |
| | **AgentExpt (Full)** | **0.4523** | **0.6938** |
| | BM25 + GPT-4o | 0.1906 | 0.4514 |
| Dataset | AgentExpt (w/o Finetune) | 0.3001 | 0.5213 |
| | **AgentExpt (Full)** | **0.3001** | **0.5549** |

**Analysis.** The results reveal two critical insights regarding the architectural design of AgentExpt:

**1. The Ceiling Effect of Retrieval.** Comparing BM25 + GPT-4o with AgentExpt(w/o Finetune), we observe a substantial performance gap. Even with the powerful GPT-4o as a reranker, the system using BM25 achieves lower HitRate@10 (0.5427 vs. 0.6501 for baselines). This is primarily because the reranker is bound by the recall quality of the retrieval stage (R@20: 0.2971 vs. 0.4523). A lexical retriever fails to capture the semantic and experimental nuance of research ideas, filtering out relevant candidates before the LLM can ever assess them. This confirms that our Collective Perception-Enhanced Retriever provides a necessary foundation that cannot be replaced solely by a strong downstream reasoner.

**2. The Value of SFT.** Comparing AgentExpt(w/o Finetune) with the full AgentExpt model highlights the specific benefit of our training strategy. While the GPT-4o performs respectably, finetuning the model on interaction chains yields consistent improvements in precision (Baseline HR@10 increases from 0.6501 to 0.6938; Dataset HR@10 from 0.5213 to 0.5549). This demonstrates that general reasoning capabilities are insufficient for the specific task of experimental design; the model benefits explicitly from learning the latent co-usage patterns embedded in our interaction chains, allowing it to better prioritize candidates that are experimentally suitable rather than merely plausible.

### A.3. Latency Analysis

To demonstrate the practicality of AgentExpt, we analyze the trade-off between recommendation performance and inference latency. Table A.9 presents the Recall@20, HitRate@10, and average per-query latency on a single NVIDIA A800 GPU.

As shown in the table, the lexical method BM25 is computationally negligible (0.005s) but fails to capture deep semantic relevance, resulting in the lowest Recall@20 (0.2971). Among neural baselines, SciBERT achieves the lowest latency (0.129s), yet its performance remains suboptimal (0.5965 HR@10) due to the lack of specialized reasoning structures. While Textual-GCL operates at a latency (0.301s) comparable to our method, it significantly lags in retrieval quality (-12.4% in HR@10 compared to ours). The strongest prior baseline, SymTax, yields competitive accuracy but incurs the highest latency of 0.692s.

*Table A.9.* Performance and latency comparison on the Paper-Baseline recommendation task. Latency is measured per query on a single A800 GPU. AgentExpt achieves the best trade-off between accuracy and efficiency.

| Method | R@20 | HR@10 | Latency (s) |
|--------|------|-------|-------------|
| BM25 | 0.2971 | 0.5171 | 0.005 |
| Textual-GCL | 0.3594 | 0.5696 | 0.301 |
| SciBERT | 0.3905 | 0.5965 | 0.129 |
| HAtten | 0.4168 | 0.6216 | 0.208 |
| SymTax | 0.4218 | 0.6453 | 0.692 |
| **AgentExpt** | **0.4523** | **0.6938** | **0.367** |

AgentExpt strikes the best balance, delivering state-of-the-art performance (0.6938 HR@10) with a highly efficient inference speed of 0.367s. From a practical usability perspective, this latency is negligible. According to foundational research in human-computer interaction, response times under 1.0 second are critical to maintaining a user's uninterrupted flow of thought (Miller, 1968; Nielsen, 1994). Since AgentExpt's latency (0.367s) is well below this threshold, the delay is effectively imperceptible during complex research workflows. Researchers can receive immediate feedback on experimental designs without experiencing cognitive breaks, confirming that our framework is ideal for real-time, interactive AI scientist agents.

## A.4. Significance Analysis

To verify that the performance gains of AgentExpt are statistically significant and not artifacts of random variation, we conducted a rigorous statistical analysis. We repeated the evaluation for both the Paper–Baseline and Paper–Dataset recommendation tasks over five independent runs using different random seeds.

Table A.10 reports the mean and standard deviation ($\mu \pm \sigma$) for Recall@20 and HitRate@10. We performed a paired $t$-test to determine the statistical significance of the improvements. The results demonstrate that AgentExpt consistently outperforms SymTax with low variance across all metrics. Specifically, we observe statistically significant improvements at the $p < 0.01$ level for Baseline Recall@20 and both Dataset metrics. Notably, for the Paper–Baseline HitRate@10, the improvement is highly significant with $p < 0.001$. These results confirm the robustness and reliability of our proposed framework.

*Table A.10.* Statistical significance analysis over 5 independent runs. We compare the strongest baseline (SymTax) with AgentExpt. Results are reported as Mean $\pm$ Standard Deviation. Statistical significance is marked as $^{**}$ for $p < 0.01$ and $^{***}$ for $p < 0.001$.

| Task | Method | R@20 | HR@10 |
|------|--------|------|-------|
| Baseline | SymTax | $0.4218 \pm 0.0132$ | $0.6453 \pm 0.0151$ |
| | **AgentExpt** | $\mathbf{0.4523 \pm 0.0074}^{**}$ | $\mathbf{0.6938 \pm 0.0120}^{***}$ |
| Dataset | SymTax | $0.2873 \pm 0.0046$ | $0.5125 \pm 0.0098$ |
| | **AgentExpt** | $\mathbf{0.3001 \pm 0.0067}^{**}$ | $\mathbf{0.5549 \pm 0.0133}^{**}$ |

## A.5. Interaction Chain Analysis

To validate the hypothesis that baselines and datasets exhibit strong co-usage dependencies—and to justify the design of our reasoning-augmented reranker—we analyze the quality of candidates generated solely through interaction chains compared to other heuristic retrieval strategies.

**Setup.** We compare three candidate generation strategies to retrieve a pool of top-100 items for a given target paper:

1. **Chain-derived:** Candidates reached via 2-hop interaction chains (e.g., Target Paper → Reference Dataset → Intermediate Paper → Candidate Baseline).

2. **Same Conference:** Candidates extracted from other papers published in the same venue and year (a temporal-community heuristic).

3. **Embedding:** Candidates with the highest cosine similarity based on the initial dense retrieval (a semantic heuristic).

We measure **Recall** (the fraction of ground-truth items covered by the pool) and the **Pool Selectivity** (the size of the retrieved pool relative to the global inventory, acting as a proxy for precision/compactness).

**Analysis.** As illustrated in Figure A.6, the chain-derived strategy serves as an exceptionally strong prior for experimental design.

- **High Coverage (Recall):** The interaction chains successfully recover the majority of ground-truth components, achieving a recall of **60.14%** for baselines and **78.61%** for datasets. This significantly outperforms the "Same Conference" heuristic, confirming that experimental choices are driven more by methodological compatibility (usage chains) than by venue proximity.

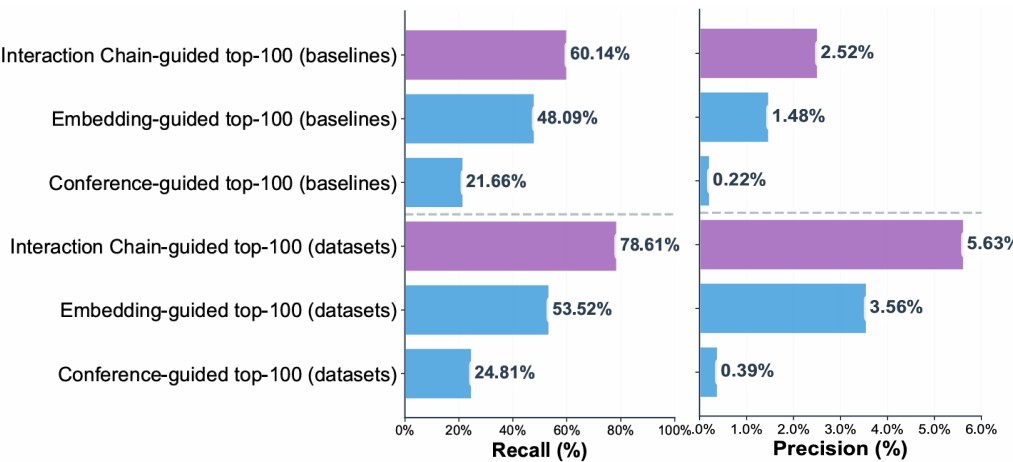

*Figure A.6.* **Constructing chain-derived candidates and analysis. Left:** From the interaction graph of papers, baselines, and datasets, we extract interaction chains and aggregate the terminal items to form a chain-derived dataset/baseline pool. **Right:** (i) *Recall (%)* between a target paper's actual baselines/datasets and candidates from each setting, and (ii) *Precision (%)* of overlapped items within the corresponding candidate pool. We evaluate three settings on both the baseline side and the dataset side: chain-derived top-100, same conference-derived top-100, and embedding top-100. Chain-derived candidates recover on average 60.14% of baselines and 78.61% of datasets while occupying only 2.52% and 5.63% of the respective total pools, indicating that interaction chains provide a compact yet highly informative prior.

- **High Density (Precision):** Crucially, the chain-derived pool achieves this high recall while remaining extremely compact. As shown in the precision metric, the chain-derived candidates occupy only **2.52%** and **5.63%** of the total available pool for baselines and datasets, respectively.

**Conclusion.** These results demonstrate that the interaction graph effectively filters out the vast majority of irrelevant candidates while preserving the correct ones. Unlike embedding-based retrieval, which relies on semantic surface forms, interaction chains leverage the latent "collaborative filtering" signal of the scientific community. This validates our design choice in the Reranker module: by feeding these high-recall, high-precision chains as reasoning evidence to the LLM, we allow the model to focus on verifying the suitability of a small, highly plausible set of candidates rather than searching blindly through the entire knowledge base.

### A.6. Query Representation Prompt

This prompt is used to transform the raw abstract of a paper into a concise research synopsis $q$, focusing on the problem setting and evaluation intent (Section 3.2).

---

**Query Representation Prompt**

```
You are an expert AI researcher assistant. Your task is to distill a research paper's abstract into a
    concise "Research Synopsis" to serve as a query for retrieving suitable baselines and datasets.

Input Abstract:
{{paper_abstract}}

Instructions:
1. Identify the core Research Problem (e.g., long-tail recognition, offline RL).
2. Identify the specific constraints, assumptions, or settings (e.g., noise-free, few-shot, multi-agent).
3. Identify the Evaluation Intent (what claims need to be tested?).
4. Remove generic background, marketing fluff, or results summaries.
5. The output must be a single, dense paragraph describing the experimental context.

Output format:
Synopsis: [Your summary here]
```

---

## A.7. Collective Perception Prompt

This prompt is used to synthesize the Collective Perception (CP) of a baseline or dataset by aggregating its citation contexts from downstream papers (Section 5.1).

---

**Collective Perception Synthesis Prompt**

```
You are a senior scientific analyst. You are provided with a list of "Citation Contexts" extracted from
    various papers that cite a specific target artifact (Baseline or Dataset): "{{target_name}}".

Your goal is to synthesize a "Collective Perception" summary that describes HOW and WHY the community uses
    this artifact.

Citation Contexts:
{{list_of_citation_contexts}}

Instructions:
1. Ignore the artifact's self-description. Focus only on the third-person evidence provided in the contexts.
2. Identify recurring usage patterns:
    - For Datasets: What tasks is it standard for? What metrics are used? Is it used for pre-training or
      evaluation?
    - For Baselines: What is it compared against? Under what settings (e.g., strong baseline for small data)
      does it excel or fail?
3. Discard generic mentions (e.g., "we compare with [X]"). Keep informative contexts (e.g., "we use [X] as
    a strong baseline for OOD generalization").
4. Synthesize this into a factual, dense profile.

Output:
Collective Perception: [Summary]
```

---

## A.8. Entity Filtering Prompt

This prompt is used in the data construction pipeline to distinguish valid experimental components from generic terms and normalize their names (Section 4.2).

---

**Entity Filtering and Normalization Prompt**

```
You are a data curator for a machine learning knowledge base.
You are given a snippet of text from a paper's experiment section and a candidate entity name extracted
    from it.

Text Snippet: "...{{text_snippet}}..."
Candidate Entity: "{{candidate_name}}"

Task:
Determine if the Candidate Entity is a concrete (1) Baseline Method or (2) Dataset used for evaluation in
    this context.
- Reject generic concepts (e.g., "Deep Learning", "Accuracy", "Parameters").
- Reject software libraries unless used as a baseline (e.g., reject "PyTorch", "Numpy").
- If valid, provide the Canonical Name (e.g., map "ResNet" -> "ResNet-50" if clear, or "ImageNet" ->
    "ILSVRC-2012").

Output JSON:
{
  "is_valid": true/false,
  "type": "baseline" or "dataset" or "n/a",
  "canonical_name": "string",
  "reasoning": "brief explanation"
}
```

---

## A.9. Reasoning Chain Generation Prompt

This prompt is used to generate the reasoning trace $A$ for the reranker. It interprets the interaction chains (Paper-Dataset-Paper-Baseline) to justify the ranking (Section 5.2).

## Reasoning Chain Generation Prompt

```
You are an expert AI Experiment Design consultant.
User Query (Research Idea): "{{query_synopsis}}"

I have retrieved a set of candidate [Baselines/Datasets]. For each candidate, I have extracted "Interaction
    Chains" from the literature. An interaction chain connects the User's idea to the candidate via
    intermediate papers and shared resources.

Format of Chain: Query_Paper -> uses -> [Intermediate_Resource] -> used_by -> [Reference_Paper] -> uses ->
    [Candidate]
(Higher frequency of chains implies higher community consensus).

Candidate Evidence:
{{candidate_evidence_list}}
(Each entry contains: Candidate Name, Top-3 Interaction Chains, Co-usage Count)

Task:
1. Analyze the interaction chains. Do the intermediate resources (datasets/baselines) align with the
    specific constraints of the User Query?
2. Prioritize candidates that share "experimental logic" with the query, not just keyword overlap.
3. Select the top-10 most suitable candidates.
4. Write a reasoning chain justifying why it fits, explicitly referencing the interaction path.

Output Format:
Ranked List: [Item 1, Item 2, ...]
Reasoning Trace: "I recommend [Item 1] because it is frequently used in conjunction with [Intermediate
    Resource], which matches the user's setting of [Constraint]. specifically, [Reference Paper]
    demonstrates this combination is effective for..."
```

