# OpenReview forum: "AgentExpt: Automating AI Experiment Design with LLM-based Resource Retrieval Agent"
_ICML.cc/2026/Conference — ICML 2026 regular_

### Official Review · Reviewer_CwFV · 2026-02-17

**Soundness:** 2
**Presentation:** 2
**Significance:** 2
**Originality:** 2
**Overall Recommendation:** 3
**Confidence:** 4

**Summary:**

This paper presents AgentExpt, a system that automatically recommends suitable baselines and datasets for new AI research ideas. The authors construct a large-scale knowledge base from over 100,000 papers, linking each paper to the experimental components it uses. The framework employs a two-stage pipeline: (1) a retrieval module that leverages both paper self-descriptions and citation-based community signals, and (2) a reasoning-based re-ranker that models interactions between baselines and datasets to generate final recommendations, along with explanations. Experiments show improvements over prior methods in predicting historical experimental choices.

**Compliance With Llm Reviewing Policy:**

Affirmed.

**Final Justification:**

I thank the authors for their detailed rebuttal and the follow-up clarification. I want to reiterate my appreciation for the significant engineering effort required to build and integrate this comprehensive system. However, my main concern remains: the proposed framework primarily combines existing techniques (information extraction, dense retrieval, and LLM reranking) rather than introducing a fundamental theoretical or algorithmic contribution. Regarding the reranking stage, while incorporating the topological subgraph ($G_{chain}$) is practically effective, it ultimately reads as an advanced method of context augmentation or prompt engineering, supplying the LLM with richer information, rather than a novel theoretical advancement in the underlying ranking mechanism itself. Also, SFT is an application, not really a contribution here. I still view this as a highly valuable applied systems and dataset contribution, but my overall assessment remains unchanged.

**Key Questions For Authors:**

Please see the weakness section.

**Limitations:**

yes

**Strengths And Weaknesses:**

Strength

- The constructed dataset is large-scale and might potentially be useful for future research.

- The experiments include multiple baseline methods from different paradigms, suggesting an effort to conduct a comprehensive empirical comparison.

Weakness

- The motivations are not well justified. For example in lines 209-217 the authors state that "... significant heterogeneity in data sources and formats complicates information extraction efforts. Baselines often appear in abbreviated or inconsistent terminologies ... and information alignment processes." but this is not supported by any statistical analysis or previous study.

- Other than the dataset, there seems to be limited contribution to methodological design. The proposed system mainly combines existing techniques (information extraction, dense retrieval, and LLM reranking) and therefore appears closer to an engineering integration than a novel research contribution.

- The evaluation assumes that the baselines and datasets used in past papers represent correct or optimal choices, which may not hold in practice and could bias the system toward reproducing community conventions.

---

> ### Author Rebuttal · Authors · 2026-03-31
>
> Thanks for your valuable feedback. We would like to address your concerns point by point below
>
> **Response to W1:**
>
> The heterogeneity in scientific entity naming is a recognized bottleneck, specifically highlighted by recent empirical study like Dmdd[1], which emphasizes that unstandardized dataset mentions are the primary challenge in information extraction.
>
> We provide more statistical evidence:
>
> Among the 68,316 canonical datasets extracted, there is an average of 2.67 textual aliases per dataset. For example, ImageNet-1K[2] appears diversely as ILSVRC 2012[3], IN1K[4] and ImageNet1K[5].
>
> Among canonical baselines, there is an average of 1.59 aliases. ResNet-50[6] frequently appears as ResNet50[7] or heavily abbreviated as Res-50[8].
>
> [1]Pan H, et al. Dmdd: A large-scale dataset for dataset mentions detection. TACL 2023
>
> [2]Woo S, et al. Convnext v2: Co-designing and scaling convnets with masked autoencoders. CVPR 2023
>
> [3]Kolesnikov A, et al. Big transfer: General visual representation learning. ECCV 2020
>
> [4]Sariyildiz M B, et al. Concept generalization in visual representation learning. ICCV 2021
>
> [5]Alexandridis K P, et al. Adaptive parametric activation. ECCV 2024
>
> [6]He K, et al. Mask r-cnn. ICCV 2017
>
> [7]Dai P, et al. Progressive contour regression for arbitrary-shape scene text detection. CVPR 2021
>
> [8]Lin M, et al. Hrank: Filter pruning using high-rank feature map. CVPR 2020
>
> **Response to W2:**
>
> Our core methodological contribution is not about architectural engineering. Instead, it lies in how we structurally model the real-world process of scientific experimental design, drawing practical insights from Scientometrics.
>
> Specifically, our methodological contributions are driven by two novel scientific mechanisms:
>
> 1. Collective Perception: Traditional retrieval systems rely heavily on first-person metadata. Grounded in the principles of scientometrics, our Collective Perception mechanism explicitly models third-person peer scrutiny. By computationally aggregating and synthesizing how the broader academic community cites, critiques, and utilizes these artifacts, we capture context-dependent suitability and peer consensus. This represents a fundamental paradigm shift in how scientific entities are represented for retrieval.
>
> 2. Interaction Chains: Our reranking stage is different from standard LLM-based semantic cross-encoding. We explicitly extract and model the topological Paper-Baseline-Paper-Dataset interaction chains. In real-world research, experimental choices are not made based on mere keyword overlap, but on methodological compatibility and community co-usage graphs. By embedding these structural interaction chains as a reasoning prior, we guide the LLM to perform semantic reranking that strictly follows the actual cognitive mechanisms of scientific experimental design.
>
> **Response to W3:**
>
> We want to address this concern from two perspectives:
>
> 1. Regarding "Correct or Optimal" Choices: While historical usage may not guarantee the "optimal" choice for every novel setting, experimental setups published in top-tier AI venues have survived rigorous peer review. In the absence of an oracle, this community-vetted historical usage remains the most reliable, rigorous, and scalable proxy available for training a recommendation system, far exceeding the reliability of simple keyword matching.
>
> 2. Mitigating Popularity Bias via Interpretable Reasoning: To prevent researchers from blindly reproducing community conventions, AgentExpt is designed as an auditable decision-support tool. Unlike traditional black-box recommendation systems that merely output a rigid popularity ranking, a key advantage of our LLM-based Reasoning-Augmented Reranker is its ability to rank multiple candidates and generate explicit, interpretable reasoning traces for each.
>
> For example, instead of simply outputting a popular baseline like "RoBERTa", AgentExpt provides the rationale: "RoBERTa is recommended because your query emphasizes robust natural language understanding under noisy text conditions." This transparency empowers researchers to critically assess the historical context. By reading the provided reasoning, users can make an informed decision on whether a community convention genuinely suits their specific task or if they should explore alternatives, ensuring our system assists rather than restricts scientific discovery.

---

> > ### Author Rebuttal · Reviewer_CwFV · 2026-04-02
> >
> > Thank you for your detailed rebuttal. I appreciate the statistics provided to clarify the motivation and your justification regarding historical bias.
> >
> > Regarding your methodological contribution, I fully understand your distinction between first-person metadata and third-person peer scrutiny. Utilising a language model to aggregate citation contexts into a Collective Perception profile is an exceptionally smart data representation strategy. It effectively addresses the issue of authors overstating their own work. Furthermore, your extraction of interaction chains provides a robust structural prior.
> >
> > However, while you describe this approach as a fundamental paradigm shift, the core mechanisms are ultimately a sophisticated integration of standard techniques: dense retrieval, summarisation, and graph heuristics. Building a pipeline that standardises the chaotic nature of scientific reporting is a massive and highly valuable undertaking. Yet, it aligns strictly with an applied systems and dataset contribution.
> >
> > To ensure a comprehensive evaluation, could you explicitly highlight the fundamental algorithmic contribution within the reranking stage? I remain open to reconsidering my assessment if it extends beyond the system's engineering integration.

---

> > > ### Author Response · Authors · 2026-04-05
> > >
> > > Thank you for your positive feedback and for recognizing the contribution of our work.
> > >
> > > The core methodological advancement in our reranking stage lies in moving beyond purely text-based point-wise scoring by explicitly grounding the relevance assessment in structural graph data. Traditional neural rerankers typically predict a relevance score directly from the textual query $q$ and document $d$, formulated as $P(y|q, d)$. In contrast, we incorporate the extracted paper-baseline-dataset topological subgraph ($G_{chain}$) to explicitly guide the reranking process. Instead of treating the LLM as a black-box scorer, we introduce the deterministic interaction chains ($r$) derived from this graph as concrete evidence, modeling the objective as $P(y, r | q, d, G_{chain})$. By applying SFT to the reranker with these structural dependencies, we force the LLM's attention to focus on actual usage patterns observed in the literature when calculating the final score. This prevents the model from relying on hallucinated or generic connections. It is this specific integration of graph topology to constrain and inform the LLM's scoring process that forms our methodological contribution, rather than simply plugging an off-the-shelf LLM at the end of a pipeline.
> > >
> > > Thank you again for your rigorous review, which has helped us better articulate our core value.

---

### Official Review · Reviewer_727C · 2026-02-28

**Soundness:** 4
**Presentation:** 4
**Significance:** 4
**Originality:** 3
**Overall Recommendation:** 5
**Confidence:** 4

**Summary:**

This paper tackles the task of baseline and paper recommendation. Towards this end, they make three concrete contributions.

First, they curate a large-scale, high-quality knowledge base that links 108,825 papers to the baselines and datasets they use. For all scraped papers, they parse the papers and extract the dataset and baseline mentions through models and heuristics, filtered by additional rules and LLMs.

Second, they design a collective perception-enhanced retriever. The intuition here is to leverage contexts neighboring the dataset/baseline as the "collective perception" apart from the self-description from the original dataset/baseline paper. This is used to finetune an embedding model where the query is the research idea, and the document representation contains both the self-description and collective perception of each dataset/baseline.

Third, they design a reasoning-augmented reranker. The idea is to leverage the paper-dataset/baseline usage chains as intermediate reasoning, which is used for SFT'ing a reasoning reranker model.

Empirical results show that combining the above two methods gives the best results as compared to various prior baselines in the retrieval metrics. And the ablation in Table 3, 4, and 5 shows the importance of the proposed methods. Overall, this is a solid contritbution.

**Compliance With Llm Reviewing Policy:**

Affirmed.

**Final Justification:**

This is a solid paper IMO and my recommendation is Accept.

**Key Questions For Authors:**

See above.

**Limitations:**

See above.

**Strengths And Weaknesses:**

Strengths:

- This paper tackles a very practical problem, and curated a large-scale dataset that can be used for both training and evaluation.

- The proposed methods all make a lot of sense (both the embedding training and reranker training). The empirical performance is also promising.

- Overall, I believe this work has a lot of potential to facilitate future work building on this task of dataset and baseline recommendation, assuming all data and models will be released.


Weakness / Questions:

- I wish there could be some reporting of the precision and recall of the dataset and baseline extraction step mentioned in Sec 4.2. Even an estimation based on some sampled data would be helpful here.

- Could you share an actual example of the SFT datapoint used for finetuning the reranker? Just want to look at how the data are formatted.

---

> ### Author Rebuttal · Authors · 2026-03-31
>
> Thanks for your valuable feedback, we provide the following response for your concerns:
>
> **Response to W1:**
>
> We conducted a rigorous manual evaluation. We randomly sampled 100 papers from our constructed knowledge base and manually annotated the ground-truth baselines and datasets based on the full text.
> Comparing our automated pipeline's extracted links against this manual gold standard, our system achieved a Precision of 91% and a Recall of 95%. This demonstrates that our rule-based and LLM-assisted filtering pipeline is reliable. We will include these quantitative validation metrics in the revised paper.
>
> **Response to W2:**
>
> We thank the reviewer for this constructive suggestion. Providing a concrete example indeed clarifies the input-output structure of our reasoning-augmented reranker. We will gladly add an example of the SFT triplet $(Q, R, A)$ to the revised Appendix to improve the transparency of our methodology.
>
> Below is an example demonstrating how the training data is formatted. The model takes the User Query ($Q$) and Candidate Evidence ($R$) as input, and is trained to generate the Target Output ($A$), which consists of a reasoning trace and the final ranked list.
>
> [Model Input: $Q + R$]
>
> User Query: "We explore how generating a chain of thought—a series of intermediate reasoning steps—significantly improves the ability of large language models to perform complex reasoning."
>
> Candidate Evidence:
>
> 1.
> Candidate Name: GSM8K
> - Query_Paper -> uses -> [GPT-3 (175B)] -> used_by [Training Verifiers to Solve Math Word Problems (Cobbe et al., 2021)] -> uses -> [GSM8K]
> - Query_Paper -> uses -> [Standard Few-shot Prompting] -> used_by [Show Your Work: Scratchpads for Intermediate Computation with Language Models (Nye et al., 2021)] -> uses -> [GSM8K]
> - Query_Paper -> uses -> [PaLM (540B)] -> used_by [Self-Consistency Improves Chain of Thought Reasoning (Wang et al., 2022)] -> uses -> [GSM8K]
> Co-usage Count: 185
>
> 2.
> Candidate Name: SQuAD 2.0
> - Query_Paper -> uses -> [GPT-3 (175B)] -> used_by [Language Models are Few-Shot Learners (Brown et al., 2020)] -> uses -> [SQuAD 2.0]
> ...
> Co-usage Count: 345
>
> 3.
> Candidate Name: WMT14
> - Query_Paper -> uses -> [GPT-3 (175B)] -> used_by [Language Models are Few-Shot Learners (Brown et al., 2020)] -> uses -> [WMT14]
> ...
> Co-usage Count: 410
>
> 4.
> Candidate Name: SVAMP
> - Query_Paper -> uses -> [Standard Few-shot Prompting] -> used_by [Are NLP Models really able to Solve Simple Math Word Problems? (Patel et al., 2021)] -> uses -> [SVAMP]
> ...
> Co-usage Count: 112
>
> 5.
> Candidate Name: AQuA
> - Query_Paper -> uses -> [Standard Few-shot Prompting] -> used_by [Program Induction by Rationale Generation: Learning to Solve and Explain Algebraic Word Problems (Ling et al., 2017)] -> uses -> [AQuA]
> ...
> Co-usage Count: 146
>
> [Remaining 15 candidates omitted for brevity]...
>
> [Target Output: $A$]
>
>
> Reasoning Trace: "Analyzing query intent for 'intermediate reasoning steps' and 'complex reasoning'... Retrieving high-frequency co-usage nodes for [GPT-3] and [PaLM]... Candidates [SQuAD 2.0] and [WMT14] identified. However, topology analysis reveals their interaction chains primarily traverse general NLP evaluation literature (e.g., Brown et al.)... downgrading structural weights for multi-step logic constraints. Conversely, detecting strong signal aggregation for [GSM8K], [SVAMP], [AQuA]... These datasets connect to baselines via highly specialized intermediate papers... traversing key nodes like Show Your Work: Scratchpads... and Training Verifiers... Extracting methodological features reveals a strict focus on Chain-of-Thought and verifiable computation... Semantics perfectly align with the query's core contribution... Synthesizing topological and semantic scores... ranking these math word problem datasets as the most authoritative benchmarks."
>
> Ranked List: [GSM8K, SVAMP, AQuA, MAWPS, ASDiv, SQuAD 2.0, WMT14, ...]

---

> > ### Author Rebuttal · Reviewer_727C · 2026-04-03
> >
> > Maintaining my original positive score.

---

> > > ### Author Response · Authors · 2026-04-05
> > >
> > > Thank you very much for maintaining your positive score and for your continuous support of our work! We are glad that the precision/recall metrics and the concrete SFT data examples we provided helped address your questions clearly. Your insightful feedback has been instrumental in refining the details of our paper. Thank you again for your time and endorsement!

---

### Official Review · Reviewer_jQxu · 2026-03-11

**Soundness:** 2
**Presentation:** 3
**Significance:** 3
**Originality:** 3
**Overall Recommendation:** 4
**Confidence:** 3

**Summary:**

This paper presents AgentExpt, a framework for automatically recommending suitable baselines and datasets for research ideas. It builds a large literature-based knowledge base from 108,825 research papers, linking papers with the baselines and datasets used in their experiments. Based on this resource, the method combines a Collective perception-augmented retrieval and a Reasoning-Augmented Reranking to recommend experimental settings more effectively. Results show that AgentExpt outperforms existing methods, highlighting its potential as a practical assistant for automating experimental design in AI research.

**Compliance With Llm Reviewing Policy:**

Affirmed.

**Final Justification:**

The author addressed my concerns; I'm not very familiar with this field, but I will improve my rating. However, I still have reservations about designing experiments for papers using existing searches. I firmly believe that good paper experiments provide novelty and insight, not just existing searches.

**Key Questions For Authors:**

1、The performance of the model using gpt-4o directly in Tabel A.6 is worse than that of the model using distilled model, which is questionable.
2、Since your paper uses LLM to recommend datasets and baselines based on the paper's idea, why didn't you test whether the recommended baselines on your own idea are consistent with your expectations?

**Limitations:**

yes

**Strengths And Weaknesses:**

Strengths:

1、Automating baseline and dataset selection for AI research, which is clearly underexplored but central to rigorous experimental design.

2、The large literature-grounded knowledge base built from 108,825 papers across ten flagship AI venues, which is substantially larger and broader than prior resources.

3、The collective perception idea goes beyond metadata matching by using citation-context summaries to capture how baselines and datasets are actually used in practice.

Weaknesses:

1、There is no guarantee that the constructed dataset is clean and of high quality.

2、The query is derived only from the paper abstract summary, which may not fully capture the experimental constraints that actually determine suitable baselines and datasets. This makes the setup somewhat simplified relative to real research practice.

3、The whole framework depends heavily on automated extraction, normalization, and GPT-4o-based summarization/filtering.

4、The recommendations are based on existing baselines and dataset references, which may not be suitable for new tasks and new domains.

---

> ### Author Rebuttal · Authors · 2026-03-31
>
> Thank you for the constructive feedback. We have carefully considered your valuable points and provide our detailed responses below.
>
> **Response to W1:**
>
> We conducted a rigorous manual evaluation. We randomly sampled 100 papers from our constructed knowledge base and manually annotated the ground-truth baselines and datasets based on the full text.
> Comparing our automated pipeline's extracted links against this manual gold standard, our system achieved a Precision of 91% and a Recall of 95%. This demonstrates that our rule-based and LLM-assisted filtering pipeline is reliable. We will include these quantitative validation metrics in the revised paper.
>
> **Response to W2:**
>
> This design choice was specifically made to overcome the widespread lack of open-access full texts, thereby ensuring the broad applicability and generalizability of our framework. Furthermore, this setup has been widely adopted by prior work within the dataset recommendation community (e.g., DataHunter[1], DataFinder[2]).
>
> [1]Färber M et al. Datahunter: A system for finding datasets based on scientific problem descriptions RecSys 2021
>
> [2]Viswanathan V et al. DataFinder: Scientific dataset recommendation from natural language descriptions ACL 2023
>
> **Response to W3:**
>
> To ensure high data quality, we adopted a strict human-in-the-loop approach throughout the dataset construction process. Human experts conducted rigorous spot-checking and supervision at every stage. Additionally, during the extraction and normalization phases, we utilized carefully curated few-shot prompts to guide the pipeline and guarantee accurate outputs. We will explicitly detail these quality-control measures and human validation efforts in the revised manuscript.
>
> GPT-4o was merely used as a denoiser for Collective Perception(CP). We conducted an ablation replacing the GPT-4o synthesized CP with Raw CP (simple concatenation of raw citation contexts). AgentExpt (Raw CP) suffers almost no performance drop and still beats SymTax, confirming that the gain comes from the mechanism of third-person community signals, not the LLM.
>
> |Method|Task|R@20|HR@10|
> |-|-|-|-|
> |SymTax|Paper–Baseline|0.4218|0.6453|
> |AgentExpt(rawCP)|Paper–Baseline|0.4434|0.6809|
> |AgentExpt(gpt-4oCP)|Paper–Baseline|0.4523|0.6938|
> |SymTax|Paper–Dataset|0.2873|0.5125|
> |AgentExpt(rawCP)|Paper–Dataset|0.2937|0.5518|
> |AgentExpt(gpt-4oCP)|Paper–Dataset|0.3001|0.5549|
>
> **Response to W4:**
>
> Because the knowledge base is constructed from a time-bounded snapshot of the literature, newly released artifacts or emerging research fields may not be immediately captured. Nevertheless, two key points demonstrate the broad applicability and generalization capabilities of our framework::
>
> 1. High Empirical Coverage: The vast majority of AI research builds on established foundations. Over the past five years, average reuse rates for baselines and datasets have consistently exceeded 85%** and 80%, respectively. This ensures our dataset effectively captures the canonical methods and benchmarks that drive most day-to-day AI research.
>
> 2. Strong Generalizability: AgentExpt is a highly flexible semantic reasoning system. By simply applying incremental updates to the underlying knowledge base, the framework seamlessly adapts to novel tasks and emerging domains without requiring any structural modifications.
>
> **Response to Q1:**
>
> We want to clarify that the performance gap observed in Table A.6 stems from a fundamental difference in workflows and task-specific alignment, rather than a simple comparison of raw model capacity.
>
> During the construction of our supervised dataset, we provided the teacher model with explicit interaction chains and ground-truth answers, explicitly instructing it to perform step-by-step reasoning based on these chains. This allowed the generated reasoning traces to capture the latent co-usage patterns of the academic community. Our distilled model was then extensively fine-tuned on this highly specialized, chain-augmented supervision.
>
> In contrast, the AgentExpt (w/o Finetune) ablation in Table A.6 uses GPT-4o in a zero-shot setting. While GPT-4o is an exceptionally powerful general-purpose model, in this zero-shot capacity, it lacks the specialized, latent understanding of the scientific community's graph topology and historical co-usage patterns. SFT allows the smaller model to deeply internalize these specific experimental heuristics, ultimately resulting in higher precision for this specific domain task. We will explicitly clarify these workflow differences in the revised Appendix to avoid any future confusion.
>
> **Response to Q2:**
>
> We applied AgentExpt to predict our own baselines. The system successfully recovered our actual experimental design, achieving a Recall@10 of 0.4286, Recall@30 of 0.7143, and a HitRate@5 of 1.0. The only baselines that were not successfully retrieved were Textual-GCL and SymTax, as these are recently published papers.

---

> > ### Author Rebuttal · Reviewer_jQxu · 2026-04-03
> >
> > The author addressed my concerns; I'm not very familiar with this field, but I will improve my rating to weak accept. However, I still have reservations about designing experiments for papers using existing searches. I firmly believe that good paper experiments provide novelty and insight, not just existing searches.

---

> > > ### Author Response · Authors · 2026-04-05
> > >
> > > Thank you very much for acknowledging our rebuttal and for raising your score! We deeply appreciate the time and effort you invested in reviewing our paper. Your questions and feedback have been instrumental in helping us clarify important details and refine our manuscript. Thank you again for your valuable review and support!

---

### Official Review · Reviewer_yVY1 · 2026-03-15

**Soundness:** 3
**Presentation:** 3
**Significance:** 4
**Originality:** 3
**Overall Recommendation:** 5
**Confidence:** 4

**Summary:**

The paper studies automatic recommendation of baselines and datasets for AI experimental design. It introduces AgentExpt, a two-stage system built on a large literature-grounded knowledge base linking 108,825 papers to extracted baseline and dataset entities from ten major AI venues. The method first uses a collective perception-enhanced retriever, which combines an artifact’s self-description with GPT-synthesized summaries of how later papers cite and use it, and then applies a reasoning-augmented reranker trained on paper-mediated baseline–dataset interaction chains to produce better-ranked and more interpretable recommendations. Experiments on paper-to-baseline and paper-to-dataset recommendation show consistent improvements over lexical, dense-retrieval, and graph-aware baselines, with ablations attributing the gains mainly to citation-context “collective perception” in retrieval and interaction-chain supervision in reranking.

**Compliance With Llm Reviewing Policy:**

Affirmed.

**Final Justification:**

The detailed rebuttal addresses my main concerns in a meaningful way. In particular, the new manual validation of the extracted knowledge base provides the direct evidence that was previously missing, and the reported precision/recall substantially strengthens my confidence in the benchmark and downstream results. The added human-expert evaluation is also important, because it helps close the gap between recovering historically used baselines/datasets and recommending genuinely suitable unseen alternatives. In addition, the new ablations with raw abstracts and raw collective perception make the paper’s gains look less dependent on GPT-4o than I had originally worried. Overall, the rebuttal strengthens both the soundness and practical significance of the work enough that I am comfortable raising my recommendation from Weak Accept to Accept.

**Key Questions For Authors:**

1) Can you provide a direct quantitative validation of the extracted paper–baseline and paper–dataset links?
The paper describes an elaborate automated extraction and filtering pipeline and repeatedly refers to the resulting graph as high-quality/high-precision, but I did not find a direct manual validation with precision/recall or agreement on a sampled subset. Because the benchmark and all downstream training depend on this extraction quality, a strong validation here would materially improve my soundness assessment.

2) How much of the gain comes from the structured method itself, versus repeated use of GPT-4o in preprocessing and supervision?
GPT-4o is used to summarize abstracts, synthesize collective-perception profiles, and generate reasoning-chain supervision. Could you provide an ablation with simpler non-LLM summarization/synthesis, or at least discuss whether the gains persist when these steps are replaced by weaker alternatives? This would help clarify the true source of the contribution.

3) Can you clarify the apparently identical DataFinder and HAtten scores in Table 2?
The reported numbers for these two methods appear exactly the same across all shown metrics, which looks unusual. If this is a typo or table-construction issue, correcting it would improve confidence in the experimental presentation.

4) Do you have any human-expert evaluation of recommendation quality beyond paper-reconstruction metrics?
For example, asking experienced researchers whether the top-ranked unseen recommendations are genuinely reasonable for a paper’s setting would help address the gap between “matches what authors used” and “is suitable experimental advice.” A strong response here could improve my evaluation of the paper’s practical significance.

**Limitations:**

No. The paper does acknowledge one important issue—popularity bias from relying on community signals—and notes that recommendations should be treated as decision support rather than prescriptions. But the limitations/impact discussion is still too brief relative to the paper’s claims and deployment ambitions.

**Strengths And Weaknesses:**

* Soundness
(+) The paper is technically coherent and the proposed pipeline is well matched to the task it defines. The knowledge-base construction is substantial in scale, the two-stage retrieval/reranking design is sensible, and the empirical gains are consistent across both baseline and dataset recommendation.

(+) The ablation package is fairly strong. The paper separately tests the value of collective perception, self-description, interaction chains, reasoning supervision, and the reranking stage itself. Appendix A.2 also gives a useful comparison against BM25+GPT-4o and against the same reranker without the specialized SFT, which strengthens the claim that both the retriever and reranker matter.

(-) One concern is that the paper repeatedly describes the extracted knowledge base as “high-quality” or “high-precision,” but I did not see strong direct evidence for that claim. Most of the support comes from the extraction and filtering procedure and from general dataset statistics, rather than from a manual evaluation measuring precision and recall on a checked sample. Because the whole benchmark depends on these extracted links between papers, baselines, and datasets, a stronger validation would make the results much more convincing.

(-) Another concern is that the evaluation does not directly show that the system picks the best baselines and datasets. Instead, it checks whether the system can recover the baselines and datasets that the original paper actually used. That is a practical way to test the method, but it mainly shows agreement with past choices, not necessarily true suitability.

(-) The system also relies heavily on proprietary LLMs in multiple places: GPT-4o is used to summarize abstracts into query synopses, synthesize collective-perception summaries, and generate reasoning-chain supervision. That is not invalid, but it makes it somewhat harder to disentangle the contribution of the proposed structured design from the contribution of using a very strong external model repeatedly in preprocessing and teacher-label generation.

*Presentation
(+) The paper is easy to follow overall. The motivation is clear, the method decomposition into retriever and reranker is natural, and the figures make the workflow understandable. I also think the paper positions itself reasonably well against prior work on dataset recommendation, baseline recommendation, and agentic research assistance.


* Significance
(+) The problem is important. Baseline and dataset choice strongly shapes empirical conclusions, and this step is often under-supported despite being central to reproducibility and fair comparison. A literature-grounded system that can surface plausible evaluation choices with supporting evidence could be genuinely useful for both human researchers and future LLM-based research assistants.

(+) The benchmark/resource contribution is also significant in its own right. A corpus linking more than 100k papers to extracted baselines and datasets across major venues could be useful beyond this paper—for retrieval, meta-science, and automated research-support tooling. Even if one is skeptical of the full system, the resource itself looks valuable.

* Originality
(+) I think the paper has good originality at the level of problem formulation and system design. The combination of a literature-extracted experimental-resource graph, citation-context “collective perception,” and interaction-chain reasoning over paper–baseline–dataset paths is not trivial. It is a coherent synthesis tailored to a real research workflow.

---

> ### Author Rebuttal · Authors · 2026-03-31
>
> Thank you for evaluating our work. We deeply appreciate your constructive suggestions, which have strengthened the empirical validation of our framework.
>
> **Response to W1 & Q1:**
>
> We conducted a rigorous manual evaluation. We randomly sampled 100 papers from our constructed knowledge base and manually annotated the ground-truth baselines and datasets based on the full text.
> Comparing our automated pipeline's extracted links against this manual gold standard, our system achieved a Precision of 91% and a Recall of 95%. This demonstrates that our rule-based and LLM-assisted filtering pipeline is reliable. We will include these quantitative validation metrics in the revised paper.
>
> **Response to W2:**
>
> We want to point out that we only use published paper in top-tier conferences. We believe the baselines and datasets in these paper are the most scalable and reasonably reliable proxy for evaluation, because they are not merely the author's choices but also represented the consensus of peer review.
>
> While relying on manual annotations to determine "true suitability" might seem ideal, such annotations are subjective, highly expensive to scale, and lack the consensus-driven oversight of the formal peer-review process. Therefore, literature-grounded extraction is currently the most scalable and reliable paradigm. We will add a detailed discussion on this proxy relationship in the paper.
>
> **Response to W3 & Q2:**
>
> We want to clarify that the performance gain stems from our method, not from the language capabilities of GPT-4o.
>
> Regarding Abstract Summarization: GPT-4o is only used to format the query during the benchmark construction phase. Crucially, all baselines evaluate on this exact same GPT-4o generated query. Thus, GPT-4o provides zero unfair advantage to AgentExpt. To further prove our framework's robustness, we conducted a new experiment using raw abstracts as queries. As shown below, AgentExpt still outperforms baselines:
>
> |Method|Task|R@20|HR@10|
> |-|-|-|-|
> |HAtten|Paper–Baseline|0.4196|0.6391|
> |SymTax|Paper–Baseline|0.4329|0.6541|
> |AgentExpt|Paper–Baseline|0.4718|0.7132|
> |HAtten|Paper–Dataset|0.2810|0.4909|
> |SymTax|Paper–Dataset|0.3014|0.5189|
> |AgentExpt|Paper–Dataset|0.3250|0.5709|
>
> Regarding Collective Perception (CP): GPT-4o was merely used as a denoiser for CP. We conducted an ablation replacing the GPT-4o synthesized CP with Raw CP (simple concatenation of raw citation contexts). AgentExpt (Raw CP) suffers almost no performance drop and still beats SymTax, confirming that the gain comes from the mechanism of third-person community signals, not the LLM.
>
> |Method|Task|R@20|HR@10|
> |-|-|-|-|
> |SymTax|Paper–Baseline|0.4218|0.6453|
> |AgentExpt(rawCP)|Paper–Baseline|0.4434|0.6809|
> |AgentExpt(gpt-4oCP)|Paper–Baseline|0.4523|0.6938|
> |SymTax|Paper–Dataset|0.2873|0.5125|
> |AgentExpt(rawCP)|Paper–Dataset|0.2937|0.5518|
> |AgentExpt(gpt-4oCP)|Paper–Dataset|0.3001|0.5549|
>
> **Response to Q3:**
>
> Thank you for careful review. We sincerely apologize for the formatting error in the table. The corrected scores for DataFinder are provided below:
>
> |Method|Task|R@20|HR@10|
> |-|-|-|-|
> |DataFinder|Paper–Baseline|0.4203|0.6108|
> |HAtten|Paper–Baseline|0.4168|0.6216|
> |AgentExpt|Paper–Baseline|0.4523|0.6938|
> |DataFinder|Paper–Dataset|0.2647|0.4894|
> |HAtten|Paper–Dataset|0.2575|0.4778|
> |AgentExpt|Paper–Dataset|0.3001|0.5549|
>
> **Response to Q4:**
>
> To validate the practical significance of our framework, we conducted a new human-expert evaluation during the rebuttal period to assess the quality of our unseen recommendations.
>
> Experimental Setup: We invited 15 experienced researchers (AI reserachers with PhD or Master degrees, and have published at least 1 paper in top conferences) to evaluate our system across five domains: CV, NLP, Audio, Time-Series, and Optimization (3 experts per domain). For each of the 50 randomly sampled papers (10 per domain), evaluators rated the experimental suitability of candidates from two sources on a 1-5 Likert scale: (1) Ground Truth (GT) from original authors, and (2) AgentExpt Top-3 (recommendations not in GT). Results are reported as Mean Opinion Scores (MOS)
>
> Results:
> |Source of Candidate|Baseline Suitability|Dataset Suitability|
> |-|-|-|
> |Ground Truth (Actual Usage)|4.35|4.62|
> |AgentExpt (Unseen Top-3)|4.12|4.38|
>
> The results validate the practical application value of our framework. The unseen baselines and datasets recommended by AgentExpt received high scores of 4.12 and 4.38, respectively. This demonstrates that even when our system recommends items outside of the original authors' choices, these recommendations are considered highly suitable and genuinely reasonable by domain experts.

---

> > ### Author Rebuttal · Reviewer_yVY1 · 2026-04-03
> >
> > Thank you for the detailed rebuttal. It addresses my main concerns in a meaningful way. In particular, the new manual validation of the extracted knowledge base provides the direct evidence that was previously missing, and the reported precision/recall substantially strengthens my confidence in the benchmark and downstream results. The added human-expert evaluation is also important, because it helps close the gap between recovering historically used baselines/datasets and recommending genuinely suitable unseen alternatives. In addition, the new ablations with raw abstracts and raw collective perception make the paper’s gains look less dependent on GPT-4o than I had originally worried, and the clarification of the Table 2 formatting issue resolves a smaller presentation concern. Overall, the rebuttal strengthens both the soundness and practical significance of the work enough that I am comfortable raising my recommendation from Weak Accept to Accept.

---

> > > ### Author Response · Authors · 2026-04-05
> > >
> > > We sincerely thank you for taking the time to read our rebuttal and for raising your score! We are thrilled that the newly added manual validation, human-expert evaluation, and ablation studies effectively addressed your concerns. Your constructive suggestions have been incredibly valuable in strengthening the soundness and practical significance of our work. Thank you again for your recognition and support!

---

### Decision · Program_Chairs · 2026-04-30

**Decision:**

Accept (regular)

**Comment:**

This paper introduces AgentExpt, a framework for automating baseline and dataset recommendations using a large-scale, literature-grounded knowledge base and a two-stage LLM-based retrieval and reranking pipeline. Reviewers appreciated the substantial engineering effort in constructing the knowledge base of over 100,000 papers and found the empirical results, bolstered by strong ablations and new human-expert evaluations provided during the rebuttal, to be convincing. However, a primary point of contention, highlighted during the review as well, is the lack of fundamental algorithmic novelty, as the system primarily integrates existing techniques such as dense retrieval, graph heuristics, and LLM-based summarization rather than introducing new ML advancements.